# Rheumatoid Arthritis in Silica-Exposed Workers

**DOI:** 10.3390/ijerph182312776

**Published:** 2021-12-03

**Authors:** Young-Sun Min, Min-Gi Kim, Yeon-Soon Ahn

**Affiliations:** 1Department of Occupational and Environmental Medicine, Soonchunhyang University Cheonan Hospital, Cheonan-si 31151, Korea; mys0303@gmail.com; 2Department of Occupational and Environmental Medicine, Dankook University Cheonan Hospital, 201 Manghyang-ro, Dongnam-gu, Cheonan-si 31116, Korea; searchthing@naver.com; 3Department of Preventive Medicine and Genomic Cohort Institute, Yonsei University Wonju College of Medicine, Wonju-si 26426, Korea

**Keywords:** cohort studies, occupational exposure, rheumatoid arthritis, silica

## Abstract

Few studies have examined rheumatoid arthritis (RA) risk and severity in Korean workers exposed to silica. We compared the hospitalization risk of RA between silica-exposed workers and the general Korean population. The study cohort consisted of male workers exposed to silica who had undergone at least one silica-associated special medical examination between 1 January 2000 and 31 December 2004 (*N* = 149,948). The data were from the Korea Occupation Safety and Health Agency. RA morbidity based on hospital admission records was estimated from 2000 to 2005 using the Korea National Health Insurance Service claims data. The standardized admission ratio (SAR) was calculated by dividing the observed number of admissions in silica-exposed workers by the expected number of admissions in the general reference population. For the sum of “Seropositive rheumatoid arthritis” (M05) and “Other rheumatoid arthritis” (M06), the SAR was higher in the silica-exposed group (1.34, 95% CI 1.08–1.64). For M05, workers with <10 years of silica exposure had a significantly higher SAR (2.54, 95% CI 1.10–5.01) than the general population. More silica-exposed workers without a diagnosis of pneumoconiosis were hospitalized for RA than the general population. Our analysis reaffirms the link between silica exposure and RA and suggests that the severity of RA is increased by silica. Further studies of silica-exposed workers with longer follow-up are needed.

## 1. Introduction

Various genetic and environmental factors increase the risk of developing rheumatoid arthritis (RA) [1]. Strong risk factors are female sex, family history of RA, and occupational and environmental factors including tobacco and dust exposure [1,2]. Many epidemiological studies and meta-analyses have reported a positive association between occupational silica exposure and the risk of RA [3,4,5]. The relationship between silica exposure and immunotoxicity is complex. A recent review suggested that an imbalance of T cell types due to silica exposure causes autoantibody production and various autoimmune diseases [2,6]. However, only a few inconclusive studies have reported molecular and cellular findings related to RA [2,7]. Pneumoconiosis is defined as any lung disease caused by inhalation of organic or inorganic dust and fibers. Rheumatoid pneumoconiosis (RP) is a disease seen in patients with RA who are chronically exposed to silica and inorganic dust (such as asbestos, coal dust) [7]. It is not clear whether RA predisposes people to pneumoconiosis or affects the progression of pneumoconiosis in predisposed individuals, or whether silica causes RA.

Occupational silica exposure occurs in mines, quarries, the ceramics industry, steel production, tunnel construction, and many other environments. In Korea, workers exposed to silica undergo regular mandatory special health examinations [8]. Nevertheless, there are few studies of RA among Korean workers exposed to silica. Beyond the relevance of RA silica exposure, it is necessary to examine whether it affects the severity of RA. Therefore, we compared hospitalization rates between silica-exposed workers and the general population of Korea, using hospitalization for RA as a severity index.

## 2. Materials and Methods

### 2.1. Study Population

In Korea, all workers exposed to occupational hazards undergo mandatory and periodic special health examinations, which are the responsibility of the employer [8]. The 180 occupational hazards include crystalline silica (mineral dust), as specified in the Occupational Safety and Health Act. Since 2000, the Korea Occupation Safety and Health Agency (KOSHA) has collected workers’ health data [9]. All silica-exposed workers undergo special health examinations that include a physician interview, pulmonary function tests, and chest X-ray. The stored worker information includes the resident registration number (RRN, a unique 13-digit number given to all Koreans), sex, birth date, and the date they joined the current company. The study cohort consisted of male workers exposed to silica who had undergone at least one silica-associated special medical examination between 1 January 2000 and 31 December 2004 (*N* = 149,948). Person-years were calculated from the date of the worker’s first silica-associated special medical examination up to 31 December 2005, the date of the first hospitalization after diagnosis of RA, or death (whichever came first). The duration of silica exposure was the date of joining the current company and the end date was the same as for the person-years calculation. Multiple admissions for the same disease were treated as one admission, such that the hospitalization data pertained to a single disease diagnosis. The first admission date was designated as the event date. This was used to calculate person-years (from the date of joining the current company to the event date or end of follow-up). The duration of silica exposure included only the periods when the workers were at the current company.

In Korea, the Industrial Accident Compensation Insurance Act data represent the official statistics for occupational pneumoconiosis [10]. The pneumoconiosis cases analyzed in this study were all workers deemed to have pneumoconiosis by the Industrial Accident Compensation Insurance Act between 1 January 1998 and 31 December 2004. If a pneumoconiosis case entered the cohort before 2000, the start date was considered as 1 January 2000; for all other cases, the start date was the date on which the worker was first deemed to have pneumoconiosis. The cohort end date is the same as for silica-exposed workers.

As the reference population, 2% of the insured individuals were selected randomly, with consideration of age, using the 2000 database of the Korea National Health Insurance Service (K-NHIS; a mandatory single-payer program). Since 1989, the K-NHIS has covered all Korean residents, and nearly all hospitalized cases have been registered in the K-NHIS database [11]. All protocols were reviewed and approved by the Institutional Review Board of Dongguk University Gyeongju Hospital (110757-201503-HR-08-01).

### 2.2. Data on Rheumatoid Arthritis

RA morbidity rate, based on hospital admission records, was obtained from 2000 to 2005 by analyzing the K-NHIS claims data. The K-NHIS database contains patients’ personal RRN, admission and discharge dates, and disease diagnoses. The RRN of each participant corresponded to the RRN in the K-NHIS database, confirming their hospital admission for RA. The diagnosis of RA was based on the WHO ICD-10 criteria. The two disease codes analyzed in this study were M05 (Seropositive rheumatoid arthritis) and M06 (Other rheumatoid arthritis).

### 2.3. Statistical Methods

The standardized admission ratio (SAR) was calculated by dividing the observed number of admissions of silica-exposed or occupational pneumoconiosis workers by the expected number of admissions in the general reference population. The SAR (with 95% confidence interval, CI) was estimated based on person-years and using mortality computation software [12]. We adjusted for age group (5-year intervals). To evaluate the risk of RA with chronic silica exposure, a stratified analysis was performed based on the duration of silica exposure (<10 vs. ≥10 years) at the current workplace.

## 3. Results

Table 1 summarizes the general characteristics of the 149,948 male workers enrolled in this study (761,843 person-years); 390,644 person-years were in the <10 years follow-up group and 371,199 were in the ≥10 years follow-up group. The mean age of the silica-exposed workers was 37.0 ± 9.5 years. The largest age group was workers in their 30s (*N* = 51,185; 34.1%). The silica exposure time was ≥10 years in 87,591 (58.7%) workers and <10 years in 61,997 (41.3%). The general characteristics of the pneumoconiosis workers that we identified were date of approval for industrial accident cover and age at the time of approval.

During the 371,199 person-years for workers exposed to silica for ≥10 years, 45 were hospitalized with RA. Of the workers exposed to silica for <10 years, 50 were hospitalized with RA, out of 390,644 person-years. For the sum of “Seropositive rheumatoid arthritis” (M05) and “Other rheumatoid arthritis” (M06), the SAR was higher (1.34, 95% CI 1.08–1.64) for the silica-exposed group (Table 2). For M05, the SAR was significantly higher (2.54, 95% CI 1.10–5.01) for the workers with <10 years of silica exposure compared with the general reference population. No association was observed between workers with ≥10 years of silica exposure and the risk of RA. For all disease codes, the SAR was significantly higher for the pneumoconiosis workers than the general reference population. For the sum of M05 and M06, the SAR was significantly higher (11.26, 95% CI 7.28–16.62) for the pneumoconiosis workers compared with the general reference population.

## 4. Discussion

In addition to debilitating effects on the joints, RA has systemic consequences due to chronic inflammation [13]. Patients with signs of severe RA, such as seropositivity, erosions, and nodules, have a higher rate of hospitalization [14,15]. The treatments used for RA, such as disease control antirheumatic agents, biologics, nonsteroidal anti-inflammatory drugs, and corticosteroids, can also have serious side effects [16]. Thus, patients with RA are potentially at an increased risk of hospitalization [14]. Our results are consistent with evidence of an association between RA and occupational silica exposure. The higher hospitalization rates that we calculated suggest that silica exposure not only affects the incidence of RA, but also increases its severity. A few studies have evaluated occupational factors associated with RA severity. It has been suggested that silica exposure exacerbates existing rheumatic diseases, resulting in hospitalization or premature death [17]. However, one study found no association between agricultural or occupational inhalant exposure and RA autoantibodies or disease severity [18]. 

Although seropositivity is known as a poor prognostic factor in RA, it is uncertain whether seropositive RA has a worse disease course compared to seronegative RA [19]. Silica exposure is reported to increase the risk of seropositive RA, but was not a risk factor for seronegative RA [20]. Subsequent studies support this result [21,22]. However, recent studies continue to report an increased risk of seronegative RA with silica exposure [4,17]. In this study, the risk of hospitalization for seronegative RA was also high, implying that silica exposure is associated with seronegative RA. The association with seronegative RA was higher in pneumoconiosis workers. The etiology and causality between the pulmonary changes seen in RA and RP are still inconclusive, but the risk of autoimmune disease in the presence of pneumoconiosis is much greater than that observed for silica exposure itself [7]. Few recent studies have examined the prevalence of RP in pneumoconiosis; before 2000, the rate was around 1% for patients with pneumoconiosis [7]. Here, the prevalence of RP in pneumoconiosis was similar to past statistics from other countries, and the SAR was 12 times higher than that of the general population.

We conducted a prospective study of nearly 150,000 Korean workers occupationally exposed to silica, using medical records from mandatory special medical examinations rather than a questionnaire. One strength of our study was that it avoided the potential bias associated with retrospective self-reports by confirming silica exposure through national registry data. However, a healthy worker effect may have been in play, in that workers unfit to work would have been excluded. Although the association might have been weakened by a healthy worker effect, we nevertheless observed a higher hospitalization rate for the silica-exposed workers. In our cohort, RA cases were identified based on K-NHIS data rather than self-reported doctor diagnoses. Since RA patients mainly receive outpatient treatment, the number of RA cases might be underestimated. One article reported that diagnoses based on National Health Insurance data tend to be more accurate for severe than mild diseases; analyses of National Health Insurance data are more accurate for the inpatient than outpatient setting, while analyses of hospital data are more accurate than those based on data from clinics [23]. Therefore, analyzing SAR data has an advantage in terms of diagnostic accuracy, especially for more symptomatic patients, despite the difficulty of RA diagnosis and the large number of outpatient treatments.

Our results should be interpreted carefully. Smoking is the most powerful environmental factor predicting RA [1], and an interaction between smoking and silica has been shown to be associated with severe RA [24]. However, we were unable to control for cigarette smoking owing to a lack of information (no data from 90% of the total cohort). Potential confounding factors, such as family history and genetic factor, could not be considered, as they are not always collected during the mandatory silica-related special medical examinations. Furthermore, the subjects’ silica exposure history was incomplete, as the duration of exposure considered only the period at the current company; the subjects’ silica exposure status at previous places of employment was not known, and they might have left their current company after enrolment in our cohort. Although there were no records of work environment exposure for our study subjects, a recent study reported that, for 2% of Korean workers, the occupational exposure limit (0.05 mg/m^3^) was exceeded [25]. If exposure to silica is unavoidable, wearing personal protective equipment is the most efficient way for workers to reduce their exposure to silica. However, there was no information on the usage of personal protective equipment in our cohort. The prevalence of female patients with RA was two to three times higher than that of men. However, women were excluded from the study because of the small number of inpatients and female workers exposed to silica. The combination of anti-cyclic citrullinated peptide antibody and rheumatoid factor positivity is associated with more severe disease and worse outcomes than seronegative RA [26]. There is no standardized method for evaluating RA severity [27], and we did not obtain mean disease activity scores or determine whether disease-modifying anti-rheumatic drugs were used.

## 5. Conclusions

The silica-exposed workers without a diagnosis of pneumoconiosis had a higher RA hospitalization rate than the general population. Our analysis re-confirms the link between silica exposure and RA, while also suggesting that the severity of RA is increased by silica. Since the SAR did not increase with the duration of silica exposure, further studies with longer follow-up of silica-exposed workers are needed.

## Figures and Tables

**Table 1 ijerph-18-12776-t001:** General characteristics of the silica-exposed and pneumoconiosis workers.

	Numbers of Silica-Exposed Workers by the Duration of Silica Exposure						Pneumoconiosis Workers (8857 Person-Years)	
	Total		<10 Years (390,644 Person-Years)		≥10 Years (371,199 Person-Years)			
	Number of Workers	%	Number of Workers	%	Number of Workers	%	Number of Workers	%
Age at entrance of cohort								
<30	41,890	27.9	35,275	40.1	6615	10.7	21	0.9
30–39	51,185	34.1	26,739	30.4	24,446	39.4	203	9.1
40–49	41,148	27.4	17,651	20.1	23,497	37.9	684	30.6
≥50	15,725	10.5	8286	9.4	7439	12.0	1329	59.4
Mean ± S.D	37.0 ± 9.5		34.9 ± 10.0		40.0 ± 7.9		60.4 ± 7.8	
Age at employed at present company							Not available	
<20	9652	6.4	2504	2.8	7148	11.5		
20–29	77,325	51.6	38,502	43.8	38,823	62.6		
30–39	37,042	24.7	24,433	27.8	12,609	20.3		
40–49	18,801	12.5	15,857	18.0	2944	4.7		
≥50	7128	4.8	6655	7.6	473	0.8		
Mean ± S.D	30.7 ± 9.8		33.3 ± 10.1		26.9 ± 8.1			
Year at first employed in present company							Not available	
<1980	5090	3.4	0	0.0	5090	8.2		
1980–1989	29,477	19.7	0	0.0	29,477	47.6		
1990–1999	57,488	38.3	30,241	34.4	27,247	43.9		
≥2000	57,893	38.6	57,710	65.6	183	0.3		
Total	149,948	100.0	87,951	100.0	61,997	100.0	2237	100.0

S.D: Standard deviation.

**Table 2 ijerph-18-12776-t002:** Standardized admission ratios of silica-exposed and pneumoconiosis workers ^1^.

Diseases(ICD-10)	Silica-Exposed Workers by Duration of Silica Exposure						Pneumoconiosis Workers ^3^ (8790 Person-Years)	
	Total (761,843 Person-Years)		<10 Years (390,644 Person-Years)		≥10 Years (371,199 Person-Years)			
	Observed Number	SAR (95% CI)	Observed Number	SAR (95% CI)	Observed Number	SAR (95% CI)	Observed Number	SAR (95% CI)
Rheumatoid arthritis (M05)	13	1.81 (0.96–3.10) 1.81 (1.07–2.88) ^2^	8	2.54 (1.10–5.01)	5	1.24 (0.40–2.90)	6	12.01 (4.38–26.13)
Other rheumatoid arthritis (M06)	82	1.27 (1.01–1.57)	42	1.32 (0.95–1.79)	40	1.21 (0.86–1.65)	20	10.99 (6.71–16.97)
M05 and M06	95	1.34 (1.08–1.64)	50	1.47 (1.09–1.94)	45	1.22 (0.89–1.64)	25	11.26 (7.28–16.62)

CI: confidence interval; SAR: standardized admission ratio. ^1^ Reference: Korean general male population. ^2^ 90% CI. ^3^ Subjects aged ≥40 years were included in the analysis.

## Data Availability

Not applicable.

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
