# Peer review of "Rheumatoid Arthritis in Silica-Exposed Workers"

_ijerph, 2021, doi:10.3390/ijerph182312776_

Round 1

Reviewer 1 Report

Overall, the article addresses a very interesting and important topic regarding the effects of silica on the risk of RA. 

In the12th line – Instead of “in RA risk”, “risk of RA” would be more appropriate.

In the 31st and 32nd lines- The reference is not fully accurate, rather reference no.1 is far more relevant to the sentence.

In the line 34 - This might benefit from a reference to a more recent publication: Wrangel O, et al, Silica dust exposure increases risk for RA: A Swedish National Registry Case-control Study. JOEM 2021 Nov 1:63(11), 951-55.

In lines 35, 36, 37 – Extra references, such as reference no.2 can make the sentence stronger.

In lines 38- 39,  70-73, 182 – I think a brief description of what pneumoconiosis is, and listing the causal exposures besides silica (e.g., asbestos, coal dust) would be helpful.  

In lines 46, 47 – Instead of “Korean silica-exposed workers 46 and the general population”, I suggest “silica-exposed workers 46 and the general population of Korea”.

In lines 79, 97– the words “with consideration of age” – what is the consideration? What age ranges has been included?

In line 104 - It states that the mean age is 11.4 +/- 8.  This must be an error. 

Following Line 109 - The authors might consider adding a column to Table 1 listing the age distribution of the comparison 2% random sample of the general Korean population.

In line 125 – Additional references may be needed to make the sentence more valid.

In lines, 133-134, 137 – more discussion of the criteria for seropositive vs seronegative is needed to clarify what the meaning is of the stratification by ICD10 codes M05, M06. 

In lines, 162-163 – “Although smoking is the most powerful environmental factor predicting RA” – needs a reference. 

In lines, 171-172 - The authors statement that wearing personal protective equipment is the "most efficient way for workers to reduce their exposure to silica" is misleading, at best.  Industrial hygiene principles of the hierarchy of controls is based on the evidence that substitution of safer materials, engineering controls, safe work practices and other administrative controls, and other actions by the employer are much more effective than personal protective equipment, which, while necessary in situations where the above methods are not feasible, is the method of last resort.      Safety Management - Hazard Prevention and Control | Occupational Safety and Health Administration (osha.gov)

Author Response

Dear Reviewer #1

We are grateful for your scrupulous attention to detail and the insightful comments. We agree to your comments that current version of our manuscript should be revised according to reviewer’s comments and to add more clarity to our findings. The changes are highlighted below. We would like to express our sincere gratitude to you. Thank you.

  1. Overall, the article addresses a very interesting and important topic regarding the effects of silica on the risk of RA.

Response : We thank you for your interest in our article.

  1. In the12th line – Instead of “in RA risk”, “risk of RA” would be more appropriate.

Response : Thank you for your comment, we edited the sentence in ‘Abstract’ section, as below.

“We compared the difference risk of RA between silica-exposed~~~”

  1. In the 31st and 32nd lines- The reference is not fully accurate, rather reference no.1 is far more relevant to the sentence.

Response : It has been mentioned in several papers. Therefore, reference no. 1 article has been added.

  1. In the line 34 - This might benefit from a reference to a more recent publication: Wrangel O, et al, Silica dust exposure increases risk for RA: A Swedish National Registry Case-control Study. JOEM 2021 Nov 1:63(11), 951-55.

Response : Thank you for your comment, we added your recommended article.

  1. In lines 35, 36, 37 – Extra references, such as reference no.2 can make the sentence stronger.

Response : Thank you for your comment, we added the reference no. 2 article .

  1. In lines 38- 39, 70-73, 182 – I think a brief description of what pneumoconiosis is, and listing the causal exposures besides silica (e.g., asbestos, coal dust) would be helpful.

Response : Thank you for your comment,. we added the paragraph in ‘Introduction’ as follows.

“Pneumoconiosis is defined as any lung disease caused by inhalation of organic or inorganic dust and fibers. Rheumatoid pneumoconiosis (RP) is a disease seen in patients with RA who are chronically exposed to silica and inorganic dust (such as asbestos, coal dust).”

  1. In lines 46, 47 – Instead of “Korean silica-exposed workers 46 and the general population”, I suggest “silica-exposed workers 46 and the general population of Korea”.

Response : Thank you for your indication. We changed it as you recommended.

“silica-exposed workers and the general population of Korea”

  1. In lines 79, 97– the words “with consideration of age” – what is the consideration? What age ranges has been included?

Response : The sampling was the authority of the Korea National Health Insurance Service data manager, and the researchers were not involved. We asked the data manager of the Korea National Health Insurance Service to conduct the general population sampling in consideration of the age distribution of the silica-exposed workers (< 30, 41.890 workers; 30-39, 51,185 workers; 40-49, 41,148; ≥ 50, 15,725) in this study, and about 2% of the insured population (more than 300,000 people) were sampled from the Korea National Health Insurance Service. The RA hospitalization of silica-exposed workers was indirectly standardized using the general population (through PAMCOMP program). This sampling process was possible before Korea's Personal Information Protection Act was strengthened in 2009, and is currently not possible.

  1. It states that the mean age is 11.4 +/- 8. This must be an error.

Response : . We are sorry for the error. We changed it as follow.

“The mean age of the silica-exposed workers was 37.0 ± 9.5 years.”

  1. Line 109 - The authors might consider adding a column to Table 1 listing the age distribution of the comparison 2% random sample of the general Korean population.

Response : Thank you for your sharp comment. In exporting data and analysis results of the Korea National Health Insurance Service approved only the necessary results and was only able to bring in Table format. For the general population, only on-site analysis was approved, and detailed data was not approved due to personal information issues. Therefore, it is impossible to present an explanation of the general population of the article other than that it is a sampled from the data of the Korea National Health Insurance Service. The general population-related information has been described in the same way as in previously published articles.

  1. In line 125 – Additional references may be needed to make the sentence more valid.

Response : Thank you for your helpful comment. we added the related article.

  1. In lines, 133-134, 137 – more discussion of the criteria for seropositive vs seronegative is needed to clarify what the meaning is of the stratification by ICD10 codes M05, M06.

Response : Thank you for your helpful comment. The description of the meaning of seropositivity in the text was insufficient. We have added content and references for this.

“Although seropositivity is known as a poor prognostic factor in RA, it is uncertain whether seropositive RA have a worse disease course compared to seronegative RA in measures of disease activity [19]. Silica exposure is reported to increase the risk of seropositive RA, but was not a risk factor for seronegative RA [20]. Subsequent studies support this result [21,22]. However, recent studies continue to report an increased risk of seronegative RA with silica exposure [4,17]. In this study, the risk of hospitalization for seronegative RA was also high, implying that silica exposure is associated with seronegative RA. The association with seronegative RA was higher in pneumoconiosis workers, supporting this result.”

  1. In lines, 162-163 – “Although smoking is the most powerful environmental factor predicting RA” – needs a reference.

Response : . Thank you for your helpful comment. We added the citation number.

  1. In lines, 171-172 - The authors statement that wearing personal protective equipment is the "most efficient way for workers to reduce their exposure to silica" is misleading, at best. Industrial hygiene principles of the hierarchy of controls is based on the evidence that substitution of safer materials, engineering controls, safe work practices and other administrative controls, and other actions by the employer are much more effective than personal protective equipment, which, while necessary in situations where the above methods are not feasible, is the method of last resort.      Safety Management - Hazard Prevention and Control | Occupational Safety and Health Administration (osha.gov)

Response : . We agree to your comments. It would be better to replace it with a safer material than silica, but there is few suitable substitutes for silica in the industrial field, and as reviewer knows well, in most cases (mining, construction industry, glass, steel industry and so on), exposure is inevitable. To avoid misunderstandings, we added the following sentence:

“If exposure to silica is unavoidable, wearing personal protective equipment is the most efficient way for workers to reduce their exposure to silica. However, there was no information on the usage of personal protective equipment in our cohort.”

We hope that the explanations and revisions of our work are satisfactory.

Yours sincerely

Reviewer 2 Report

The article appears to be a worthwhile addition to the literature. I have some comments below:

  1. It is not clear why the study period is specifically between 2003-2004? 
  2. Inclusion and exclusion criteria should be specified. There is not enough information on how the authors recruited the reference group.
  3. Why the authors have selected 10 years cut-off on the duration of silica exposure? You could probably check the results by increasing of one-year exposure?
  4. Table 1 should be organized, the total % in age at the entrance of the cohort isn’t correct. Please check others.
  5. Line 104, I cannot find the mean age of 11.4 in your results.
  6. Line 163, is there any reason why the authors haven’t adjusted the models for smoking while it is a potential confounding factor predicting RA?  Also, there are many other potentially important confounders, but the authors haven't considered these variables.
  7. More discussion on the biological mechanisms between preconception exposure and RA is warranted. More results from previous studies should be compared and discussed in the discussion. A justification on a recommendation to reduce the exposure or mitigate the risk is recommended. 

Author Response

Dear Reviewer #2

We are grateful for your scrupulous attention to detail and the insightful comments. We agree to your comments that current version of our manuscript should be revised according to reviewer’s comments and to add more clarity to our findings. The changes are highlighted below. We would like to express our sincere gratitude to you. Thank you.

  1. It is not clear why the study period is specifically between 2003-2004?

Response : The Korea National Health Insurance Service's hospitalization records were only available after 3-4 years to be verified electronically. The time when the electronic records were checked for this study was in 2009, and at this time, it was possible to check them until 2005. This process was possible before Korea's Personal Information Protection Act was strengthened in 2009, and is currently not possible.

  1. Inclusion and exclusion criteria should be specified. There is not enough information on how the authors recruited the reference group.

Response : The sampling was the authority of the Korea National Health Insurance Service data manager, and the researchers were not involved. We asked the data manager of the Korea National Health Insurance Service to conduct the general population sampling in consideration of the age distribution of the silica-exposed workers (< 30, 41.890 workers; 30-39, 51,185 workers; 40-49, 41,148; ≥ 50, 15,725) in this study, and about 2% of the insured population (more than 300,000 people) were sampled from the Korea National Health Insurance Service. The RA hospitalization of silica-exposed workers was indirectly standardized using the general population (through PAMCOMP program). This sampling process was possible before Korea's Personal Information Protection Act was strengthened in 2009, and is currently not possible.

In exporting data and analysis results of the Korea National Health Insurance Service approved only the necessary results and was only able to bring in Table format. For the general population, only on-site analysis was approved, and detailed data was not approved due to personal information issues. Therefore, it is impossible to present an explanation of the general population of the article other than that it is a sampled from the data of the Korea National Health Insurance Service. The general population-related information has been described in the same way as in previously published articles.

  1. Why the authors have selected 10 years cut-off on the duration of silica exposure? You could probably check the results by increasing of one-year exposure?

Response : Thank you for your comment. We tried to classify it into various periods such as 5 year cut-off and 10 year cut-off, but we could not observe the trend according to the exposure period. Therefore, we presented using the same 10-year cut-off as in our previously published hazard follow-up studies (using the same data set).

  1. Table 1 should be organized, the total % in age at the entrance of the cohort isn’t correct. Please check others.

Response : We are very sorry to have caused you trouble. The incorrect number of subjects in the Table 1 is corrected after rechecking.

  1. Line 104, I cannot find the mean age of 11.4 in your results.

Response : We are sorry for the error. We changed it as follow.

“The mean age of the silica-exposed workers was 37.0 ± 9.5 years.”

  1. Line 163, is there any reason why the authors haven’t adjusted the models for smoking while it is a potential confounding factor predicting RA? Also, there are many other potentially important confounders, but the authors haven't considered these variables.

Response : Thank you for your sharp comment. Smoking is known as one of the most significant environmental risk factors for RA. Although the smoking rate, which was recorded for only 10% of our cohort, did not differ significantly in the silica‐exposed workers and the general population, we were unable to control for cigarette smoking owing to a lack of information. We edited the sentence in ‘Discussion’ section, as below.

“Smoking is the most powerful environmental factor predicting RA [1], an interaction between smoking and silica has been shown to be associated with severe RA [24]. However, we were unable to control for cigarette smoking owing to a lack of information (no data from 90% of the total cohort).”

  1. More discussion on the biological mechanisms between preconception exposure and RA is warranted. More results from previous studies should be compared and discussed in the discussion. A justification on a recommendation to reduce the exposure or mitigate the risk is recommended.

Response : Thank you for your comment. As pointed out by the reviewer, additional considerations have been added to the “Discussion” section.

“Although seropositivity is known as a poor prognostic factor in RA, it is uncertain whether seropositive RA have a worse disease course compared to seronegative RA in measures of disease activity [19]. Silica exposure is reported to increase the risk of seropositive RA, but was not a risk factor for seronegative RA [20]. Subsequent studies support this result [21,22]. However, recent studies continue to report an increased risk of seronegative RA with silica exposure [4,17]. In this study, the risk of hospitalization for seronegative RA was also high, implying that silica exposure is associated with seronegative RA. The association with seronegative RA was higher in pneumoconiosis workers, supporting this result.”

“Although smoking is the most powerful environmental factor predicting RA [1], we were unable to control for cigarette smoking owing to a lack of information (no data from 90% of the total cohort).”

“If exposure to silica is unavoidable, wearing personal protective equipment is the most efficient way for workers to reduce their exposure to silica. However, there was no information on the usage of personal protective equipment in our cohort.”

We hope that the explanations and revisions of our work are satisfactory.

Yours sincerely

Round 2

Reviewer 2 Report

The manuscript has been modified according to the points and it can be now accepted.